

# Comprehensive transcriptional profiling of aging porcine liver

Jianning Chen[1,*], Qin Zou[1,*], Daojun Lv[2], Muhammad Ali Raza[3], Xue Wang[1], Yan Chen[1], Xiaoyu Xi[1], Peilin Li[2], Anxiang Wen[1], Li Zhu[4], Guoqing Tang[4], Mingzhou Li[4], Xuewei Li[4] and Yanzhi Jiang[1]

[1] Department of Zoology, College of Life Science, Sichuan Agricultural University, Ya'an, Sichuan, China
[2] Sichuan Weimu Modern Agricultural Science and Technology Co., Ltd, Chengdu, Sichuan, China
[3] Department of Crop Cultivation and Farming System, College of Agronomy, Sichuan Agricultural University, Chengdu, Sichuan, China
[4] Farm Animal Genetic Resources Exploration and Innovation Key Laboratory of Sichuan Province, Sichuan Agricultural University, Chengdu, Sichuan, China
* These authors contributed equally to this work.

Corresponding author
Yanzhi Jiang, jiangyz04@163.com

## ABSTRACT

**Background**. Aging is a major risk factor for the development of many diseases, and the liver, as the most important metabolic organ, is significantly affected by aging. It has been shown that the liver weight tends to increase in rodents and decrease in humans with age. Pigs have a genomic structure, with physiological as well as biochemical features that are similar to those of humans, and have therefore been used as a valuable model for studying human diseases. The molecular mechanisms of the liver aging of large mammals on a comprehensive transcriptional level remain poorly understood. The pig is an ideal model animal to clearly and fully understand the molecular mechanism underlying human liver aging.

**Methods**. In this study, four healthy female Yana pigs (an indigenous Chinese breed) were investigated: two young sows (180-days-old) and two old sows (8-years-old). High throughput RNA sequencing was performed to evaluate the expression profiles of messenger RNA, long non-coding RNAs, micro RNAs, and circular RNAs during the porcine liver aging process. Gene Ontology (GO) analysis was performed to investigate the biological functions of age-related genes.

**Results**. A number of age-related genes were identified in the porcine liver. GO annotation showed that up-regulated genes were mainly related to immune response, while the down-regulated genes were mainly related to metabolism. Moreover, several lncRNAs and their target genes were also found to be differentially expressed during liver aging. In addition, the multi-group cooperative control relationships and constructed circRNA-miRNA co-expression networks were assessed during liver aging.

**Conclusions**. Numerous age-related genes were identified and circRNA-miRNA co-expression networks that are active during porcine liver aging were constructed. These findings contribute to the understanding of the transcriptional foundations of liver aging and also provide further references that clarify human liver aging at the molecular level.

## INTRODUCTION

The aging process leads to the gradual loss of the ability of an organism to maintain homeostasis and its structure alters. The organism therefore becomes more vulnerable to external stresses or damage (*López-Otín et al., 2013*). Aging significantly affects a number of physiological and biochemical functions that are performed by the liver (*Kitani, 1994*). Aging has been suggested as a major risk factor for the development of various diseases (*Rodwell et al., 2002*), and several studies showed that aging increased the risks for numerous liver diseases that cause an increase in the mortality rate (*Amarapurkar et al., 2007*; *Regev & Schiff, 2001*; *Sheedfar et al., 2013*). Previous studies have shown that the liver size progressively decreased in response to aging and a marked decline in volume was found to occur above the age of 60 years in humans (*Woodhouse & James, 1990*). This has been suggested to be related to an age-dependent decrease in hepatic blood flow (*Wynne et al., 2010*; *Zoli et al., 1999*). The aging liver in humans has been characterized to have macrohepatocytes, polyploidy, and increased numbers of in nuclei and nucleoli (*Schmucker, 1998*). It has been reported that the rate of hepatic polyploidization accelerated after five decades of life in humans (*Kudryavtsev et al., 1993*). In the liver, the neural fat and cholesterol volumes gradually expand in response to ageing, and this accumulation of lipids may damage the normal liver function by promoting organ-specific toxic reactions (*Kim, Kisseleva & Brenner, 2015*; *Slawik & Vidal-Puig, 2006*). More importantly, the aging process in the liver can lead to a decreased number and dysfunction of mitochondria, as well as a decrease in the area of the smooth endoplasmic reticulum. These effects decrease the generation of smooth endoplasmic reticulum and reduce the synthesis of microsomal proteins (*Kim, Kisseleva & Brenner, 2015*). The changes of structure and dysfunction in the liver in response to age were considered to be associated with a significant impairment of many hepatic metabolic and detoxification activities (*Bertolotti et al., 2014*).

Due to the continuous development of molecular biology, the aging process is currently understood not only a process accompanied by morphological changes, but also as a process that involves complicated changes at the molecular level. Recently, several studies have shown that DNA methylation, as an important part of epigenetics, changes during human liver aging (*Bacalini et al., 2018*; *Bysani et al., 2016*; *Huse et al., 2015*). A number of DNA methylation changes have been found to be associated with gene expression during aging in the liver (*Bysani et al., 2016*). Moreover, numerous gene expression changes have been reported during aging in the mouse liver (*Kwekel et al., 2010*; *White et al., 2015*), and age-related genes have been reported to be involved in immune response, metabolic processes, cell activation, and RNA modification (*White et al., 2015*). In addition, the non-coding RNAs including microRNAs (miRNAs), long non-coding RNAs (lncRNAs), and circular RNAs (circRNAs) have been found to play an important role in the regulation of gene expression (*Chen & Yang, 2015*; *Eulalio, Huntzinger & Izaurralde, 2008*; *Mercer, Dinger & Mattick, 2009*), and to be closely related to many diseases (*Chen et al., 2013*; *Feng et al., 2014*; *Lukiw, 2013*; *Zhao et al., 2016*). Furthermore, numerous miRNAs and lncRNAs were identified to be differentially expressed during liver aging in both humans and mice (*Capri et al., 2017*; *Shima et al., 2014*; *White et al., 2015*). However, the changes involved

in the comprehensive transcriptome, including mRNA, lncRNA, miRNA, and circRNA, during liver aging remain largely unknown both in humans and other big mammals. More importantly, previous studies have shown that the liver weight tends to increase in rodents, while it decreases in humans with age (*Kitani, 1994*). Therefore, the mouse may not be a good model animal to clearly and fully understand the underlying molecular mechanisms of human liver aging. It has been reported that pigs have a genomic structure, as well as both physiological and biochemical features that are similar to those of humans. The prominent exception is their anatomic structure (*Welsh et al., 2009*), which is more similar than that of mice (*Wernersson et al., 2005*). The pig has been identified as a valuable model to study human diseases (*Lunney, 2007*; *Zhu et al., 2015*). The above-mentioned information suggests that the pig may be an ideal model animal to obtain an understanding of the aging of the human liver at the transcriptional level.

Puberty and reproductive exhaustion are the two key points in the reproductive development of pigs, and the median ages of both sow puberty and reproductive exhaustion are about 180 days and 8 years, respectively (*Jones, Rothschild & Ruvinsky, 1998*). Both of these critical time points correspond to the age of 14 and 50 in humans, respectively (*Hansen et al., 2008*). Moreover, the liver has been found to be in a recession at the age of 50 in humans, and the aging rate entered a period of accelerated decline after the age of 50 (*Kudryavtsev et al., 1993*). To better understand the comprehensive transcriptional profiling of liver aging, pigs were used as model animals and an integrated multi transcriptome-wide profiling (mRNA, miRNA, lncRNA, and circRNA) analysis was performed via high-throughput sequencing of the liver from two young (180 days) and two old (8 years) sows. This procedure enabled to identify numerous age-related genes and unclear multi-group cooperative control networks in the aging pig liver. In summary, the obtained data may help to explain age-associated changes in the transcriptional patterns during liver aging, and also provide further references toward an understanding of human liver aging at the molecular level.

## MATERIALS & METHODS

### Animals

In this experiment, four Yanan sows were used to investigate the porcine liver aging process. For this study, two young sows (180-days-old) and two old sows (8-years-old) were selected. Moreover, no direct and collateral blood relationship existed among the four pigs within the last three generations of these experimental animals. All experiments involving animals were conducted according to the Regulations for the Administration of Affairs Concerning Experimental Animals (Ministry of Science and Technology, China, revised in June 2004), and approved by the Institutional Animal Care and Use Committee in College of Animal Science and Technology, Sichuan Agricultural University (Sichuan, China) under permit No. SKY-B20160201.

Feeding and slaughtering standards of the pigs were performed as described in previous literature (*Chen et al., 2018*). As in previous research, piglet weaning was executed at the age of 28 ± 1 days. After weaning, an initial diet consisting of 3.40 Mcal kg$^{-1}$ metabolizable

energy having 20.0% crude protein (11.5 g kg$^{-1}$ lysine) was provided to piglets, which continued from day 30 to 60. The pigs were shifted to a different diet, composed of 14.0 MJ kg$^{-1}$ of metabolizable energy and 18.0% crude protein (9.0 g kg$^{-1}$ lysine) from the 61st to the 120th day. On the 121st day, pigs were subjected to a feed constituting 16.0% crude protein (8.0 g kg$^{-1}$ lysine) and 13.5 MJ kg$^{-1}$ of metabolizable energy. Similar conditions were provided to pigs with the permission to freely access food and water. Pigs were not allowed to access food one night before slaughtering and had a resting period of 2 h after transportation. To exsanguinate and relieve the pain, a sudden electric shock of 90 V and 50 Hz for 10 s was administered (*Chen et al., 2018*).

## Sample preparation

As previous study described, all samples were investigated following the rules and regulations for the handling and establishment of research animals by the Ministry of Science and Technology of China (*Chen et al., 2018*). Porcine liver tissue was harvested from each pig and frozen in liquid nitrogen. After that, samples were kept in a freezer at −80 °C until RNA extraction. The TRIzol Reagent (Invitrogen, CA, USA) was used to extract the total RNA, which was subsequently treated with DNase and purified by using the RNeasy Mini Kit (Qiagen, CA, USA). The quality and concentration of RNA were determined by an Agilent Bioanalyzer 2100 system.

## RNA sequencing

RNA samples were prepared by using about 5 μg RNA per sample. According to the manufacturer's information, the rRNA-depleted RNA by NEBNext® Ultra$^{TM}$ Directional RNA Library Prep Kit for Illumina® (New England Biolabs Ipswich, MA, USA) was used to obtain sequencing libraries. The Agilent Bioanalyzer 2100 system was utilized to analyze the library quality. After clusters had been generated, RNA sequencing was performed using Illumina HiSeq 4000 and 150 bp paired-end reads were obtained.

## Small RNA sequencing

A small RNA library was prepared by taking a total of 5 μg of RNA per sample as input material. To generate sequencing libraries, NEBNext® Multiplex Small RNA Library Prep Set for Illumina® (New England Biolabs Ipswich, MA, USA) was used, according to the manufacturer's information. To attribute sequences to each sample, index codes were supplemented. Here, the Agilent Bioanalyzer 2100 system was also utilized to test the quality of the library. According to the information provided by the manufacturer, a cBot Cluster Generation System using TruSeq SR Cluster Kit v3-cBot-HS (Illumina, Inc, CA, USA) was used to cluster the index-coded samples. After the clusters had been generated, the Illumina MiSeq platform was used to sequence the obtained library preparations, and 50 bp single-end reads were obtained.

## Statistical analysis

Clean reads of RNA sequencing were obtained from raw data after eliminating the adapter, poly-N, and low-quality reads. The clean reads were aligned to the reference genome database (Ensemble *Susscrofa 10.2*) using TopHat2 (v2.0.14) and default parameters were

used (*Kim et al., 2013*). Mapped reads per sample were assembled by using the StringTie software (*Pertea et al., 2015*), which at least presented in one of the two replicates. Mapped transcripts blasted (e-value = 1e−10) to Ensemble were directly described as known mRNA or lncRNA. To estimate the transcripts per million (TPMs) of both lncRNAs and mRNA, Salmon (v0.6.0) was utilized (*Patro, Duggal & Kingsford, 2015*). To examine the transcript coding potential, Coding Potential Calculator (CPC) (0.9) (*Kong et al., 2007*) and Pfam Scan (v1.5) (*Punta et al., 2012*) were utilized. Coding potential transcripts predicted by one or both tools were filtered out, and non-coding potential transcripts were identified as the candidate set of novel lncRNAs. The circRNAs were predicted by CIRCexplorer2 (junction reads ≥ 2) (*Zhang et al., 2016*).

The miRBase21 was used as reference and the software mirdeep2 (*Friedländer et al., 2011*) was utilized to identify the known miRNA and to predict novel miRNA. The miRanda (v3.3a) software (*Betel et al., 2008*) was used to predict the target genes of miRNAs and default parameters and cutoffs (Score S ≥ 140 and Energy E ≤ −20.0) were used. Later, the expression levels of miRNAs were also measured by TPMs.

The mRNAs, lncRNAs, miRNAs, and circRNAs were identified to be differentially expressed by using the edgeR package (*Robinson, McCarthy & Smyth, 2010*) in the R programming environment. Moreover, in the "classic analysis" section, the edgeR user guidelines were adopted in detail. A $q$-value <0.05 and fold change >2 were used to determine differentially expressed mRNAs, lncRNAs, and miRNAs in the old group vs. young group. For differentially expressed circRNAs, the standard $p$-value <0.05 and a fold change >2 was used. Here, the fold change of mRNAs, lncRNAs, miRNAs, and circRNAs was the log2 transformed ratio of the average expression between two groups, and was estimated by edgeR.

The GOseq R package was used to perform the Gene Ontology (GO) enrichment analysis, and GO terms with a corrected $p$-value <0.05 were considered to be significantly enriched by the genes. Principal Component Analysis (PCA) was performed by the prcomp function in R. All the Pearson's correlation coefficient analysis in our research used the cor function in R programming.

## CircRNA-miRNA co-expression network

The correlation analysis was the base for the construction of a circRNA-miRNA co-expression network between the differentially expressed circRNA and miRNAs. The Pearson's correlation coefficient ($r$) was used for the expression analysis of differentially expressed circRNAs and miRNAs. The $r$ >0.8 and a $p$-value <0.05 were considered relevant for network construction between a circRNA and a miRNA.

## Quantitative PCR validation

The oligo (dT) and random 6-mer primers given in the PrimeScript RT Master Mix kit (TaKaRa, Shiga, Japan) were utilized for cDNA synthesis. The CFX96 Real-Time PCR detection system (Bio-Rad Laboratories, Inc., Hercules, CA, USA) was used to conduct q-PCR using the SYBR Premix Ex Taq kit (Takara, Shiga, Japan). Three biological replicates were taken to perform the q-PCR analysis. In this assay, three endogenous control genes

(porcine *GAPDH*, *ACTB*, and *U6* snRNA) were used. The expression levels of objective mRNAs, miRNAs, lncRNAs, and circRNAs were determined by the $2^{-\Delta\Delta Ct}$ method. The Pearson's correlation coefficient of the log2 fold change values between RNA-seq and q-PCR was used to validate the reliability of the obtained RNA-seq data.

## RESULTS

### Transcriptome profile of the aging liver

To determine the changes in transcriptome level due to aging in the liver, four porcine liver tissues were obtained from two age groups. Here, RNA sequencing (RNA-seq) was used to explore transcriptome-wide profiling including mRNA, miRNA, lncRNA, and circRNA. An estimate ~71.66 million clean reads were achieved per RNA-seq library. An alignment of ~73.40–79.40% of clean reads were found with the porcine reference genome (Ensemble Susscrofa 10.2) (Table S1). Moreover, 8.00–9.43 million clean reads were gained per small RNA-seq library. 74.39–80.24% alignment of clean reads were mapped reads (Table S2). Among these four samples (including two young livers (YL) and two old livers (OL)), a total of 21,607 mRNAs, 4028 lncRNAs, 888 miRNAs, and 6366 circRNAs were detected (Fig. 1, Data S1). Out of 888 miRNAs, 336 miRNAs were identified as novel miRNAs (Fig. 1E). A total of 84.5%, 64.4%, and 95.2% of these identified mRNAs, lncRNAs, and miRNAs, respectively, were expressed in both young and old groups; however, only circRNAs had an expression of 16.9%. Moreover, the number of only expressed mRNAs, lncRNAs, miRNAs, and circRNAs in OL exceeded those only expressed in YL (Fig. 1, Data S2).

Unsupervised Euclidean matrix plots for expressed mRNA and lncRNA showed that both the old individuals and younger individuals were grouped separately (Fig. 2A), indicating an age-dependent expression pattern. Moreover, a closer distance was found between old individuals compared to that of younger individuals. PCA was used for the identified mRNA and lncRNA transcripts, and the results showed that the young group had a high degree of variance between each biological replicate, while old individuals clustered (Fig. 2B). Furthermore, four samples were investigated using unsupervised Euclidean matrix plots based on miRNAs (Fig. 2C). The result also indicated age-dependent expression patterns. However, the result of PCA for miRNA has shown that there was a higher degree of variance between each biological replicate in the old group than in the young group (Fig. 2D), which contrasted with the PCA result based on mRNA and lncRNA.

### Differentially expressed mRNAs during liver aging

Form our results, though most genes had been shared for both ages during liver development, we also detected a proportion of age-dependent differentially expressed genes (DEGs). First, DEGs were screened in OL vs. YL (OYL), which identified 273 DEGs, of which 238 were up-regulated and 35 were down-regulated (Fig. 3A, Data S3). To explore the biological functions of these age-related genes, GO analysis was performed. Compared to the young group, the results showed that the up-regulated genes were significantly enriched for immune response, the regulation of cell differentiation, stress response, both transport and storage of lipids, such as defense response to virus (GO:0051607), and ISG15-protein conjugation (GO:0032020); positive regulation was

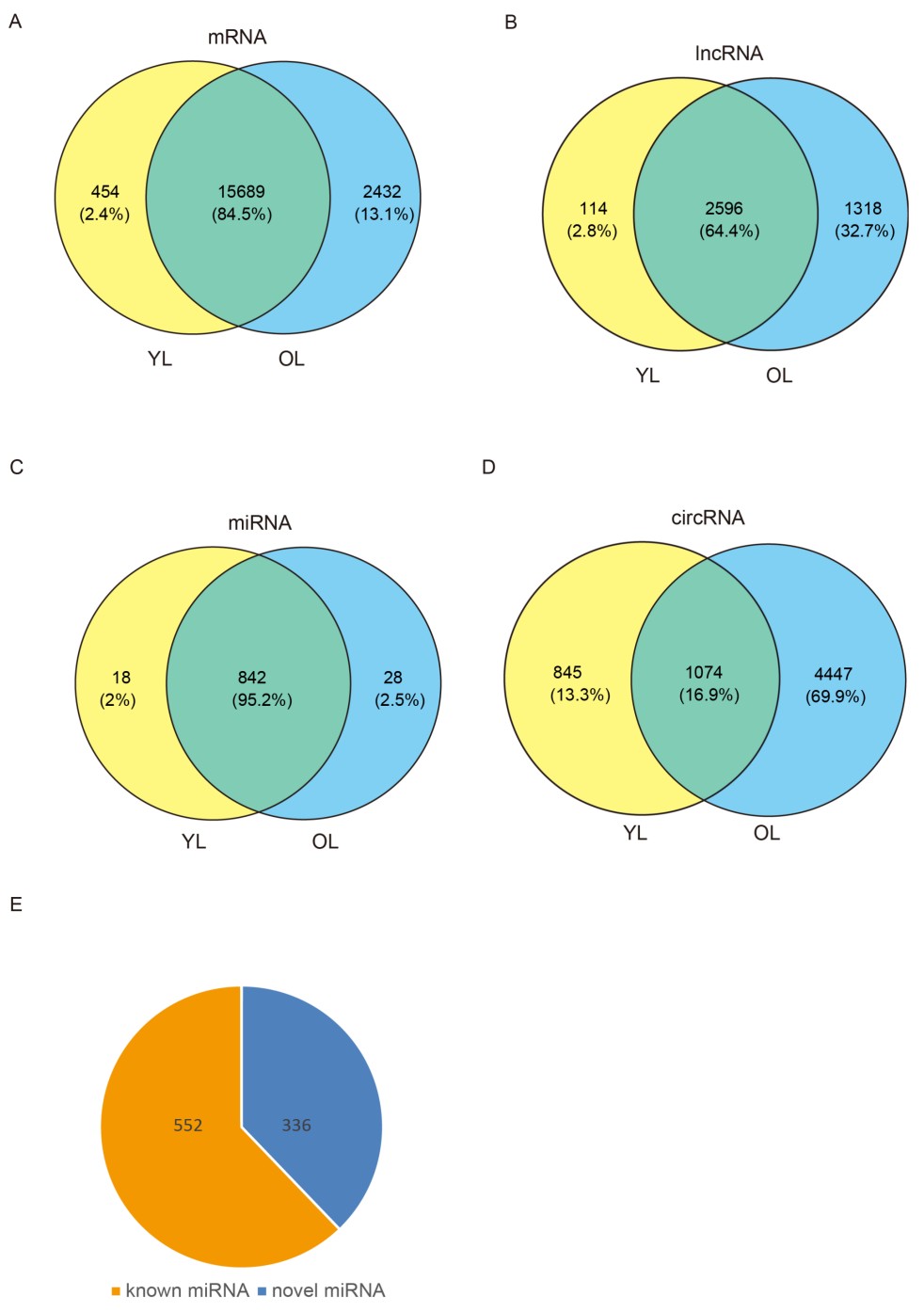

**Figure 1** **Global mRNA, lncRNA, miRNA, and circRNA expression patterns across liver samples.** (A) mRNA. (B) lncRNA. (C) miRNA. (D) circRNA. (E) Known and novel miRNAs. YL, young liver; OL, old liver.

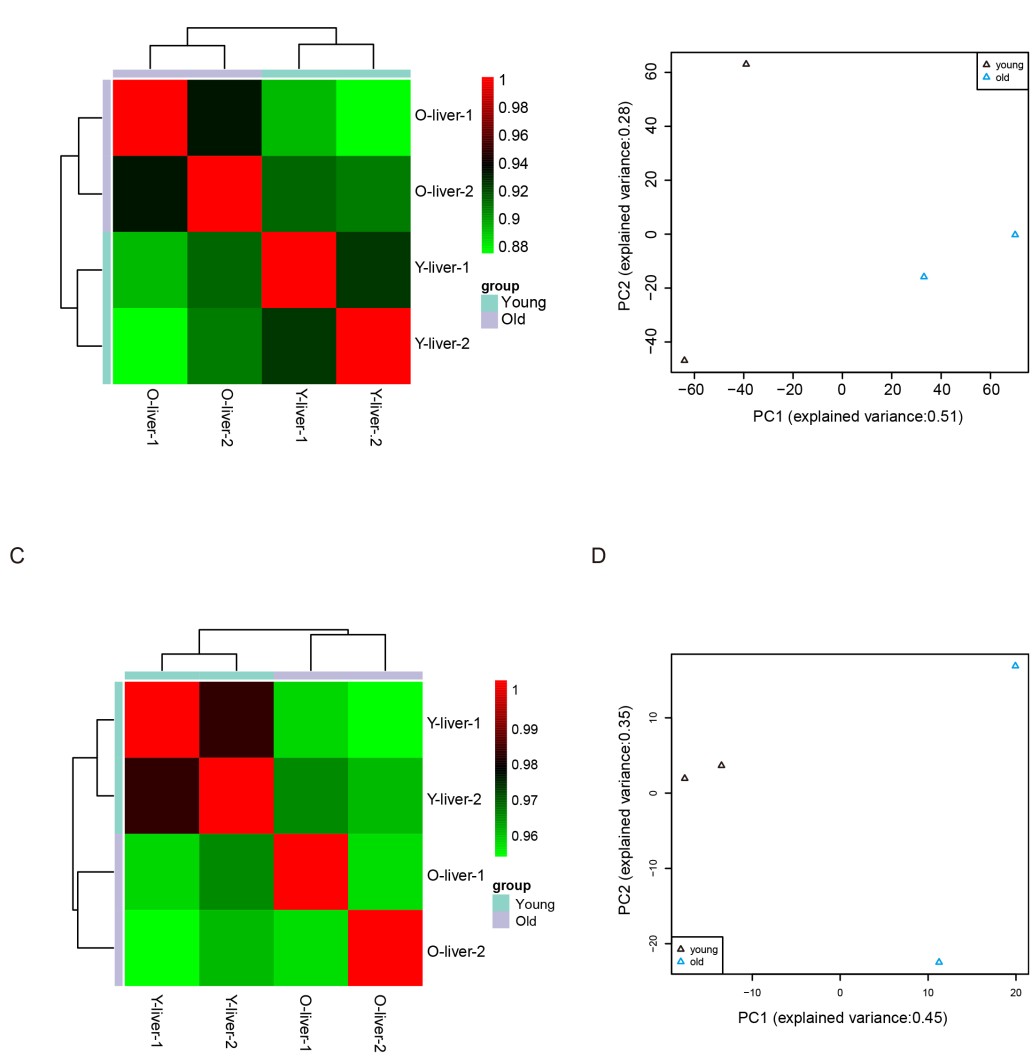

**Figure 2  Variance between old and young individuals.** (A) Pearson's correlation coefficient heat map of mRNA and lncRNA between YL and OL. The identified mRNA and lncRNA expression values (TPM) in every sample were used to perform the Pearson's correlation coefficient analysis, and values closer to 1 were less variable. (B) Principal Component Analysis (PCA) plot based on the normalized expression level (log2 (TPM)) of identified mRNAs and lncRNAs. (C) Pearson's correlation coefficient heat map of miR-NAs between YL and OL. The identified miRNA expression values (TPM) in every sample were used to perform the Pearson's correlation coefficient analysis, and values closer to 1 were less variable. (D) PCA plot based on normalized expression level (log2 (TPM)) of identified miRNAs.

found for cell differentiation (GO:0045597), response to biotic stimulus (GO:0009607), positive regulation of lipid storage (GO:0010884), and chylomicron remnant clearance (GO:0034382) (Fig. 3C, Data S4). For example, the gene nuclear receptor subfamily 1 group H member 2 (*NR1H2*, ENSSSCG00000003211) was identified to be up-regulated (log FC = 2.026481405) (Data S3) during liver aging, and it was GO enriched for the positive regulation of lipid storage (Data S4). The *NR1H2* encoding liver X receptor (LXR)-*β*

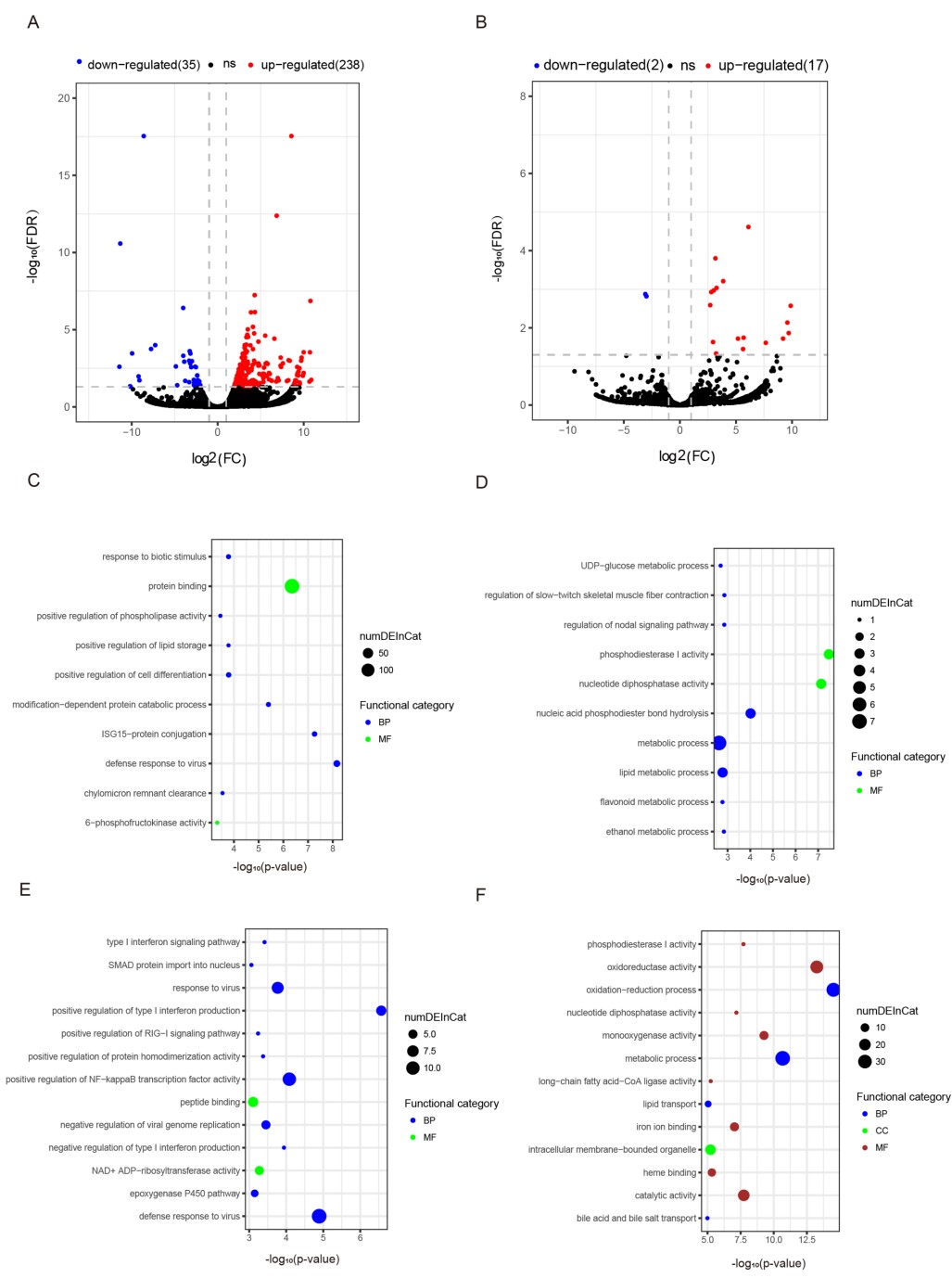

**Figure 3 Differentially expressed mRNAs and lncRNAs during aging.** The *q*-value < 0.05 and fold change >2 were used to identify differentially expressed mRNAs and lncRNAs in OYL. Gene Ontology (GO) enrichment analyses were performed to analyze the functional enrichment of differentially expressed mRNAs and lncRNAs. (A) Differentially expressed mRNAs in OYL. (B) Differentially expressed lncRNAs in OYL. (C) Functional enrichment analysis for up-regulated mRNAs in OYL. (D) Functional enrichment analysis for down-regulated mRNAs in OYL. (E) Function enrichment analysis for the target genes of up-regulated lncRNAs in OYL. (F) Function enrichment analysis for the target genes of down-regulated lncRNAs in OYL. Only the most enriched (*p* < 0.05) and meaningful GO terms are presented here. OYL, old liver vs. young liver; red dots and blue dots represent up-regulated and down-regulated mRNAs during aging, respectively. FDR, false discovery rate; FC, fold change. BP, biological process; CC, cellular component; MF, molecular function.

was found to be universally expressed (*Kalaany & Mangelsdorf, 2006*), and the variation within the *NR1H2* gene may facilitate the development of type 2 diabetes (*Ketterer et al., 2011*). Moreover, the genetic variability at *NR1H2* may contribute to an increased risk of Alzheimer's disease (*Adighibe et al., 2006*), and the preeclampsia was associated with a polymorphism in *NR1H2* (*Mouzat et al., 2011*).

The down-regulated genes were mainly related to diphosphatase activity, signaling pathway, and metabolic processes, such as phosphodiesterase I activity (GO:0004528), nucleotide diphosphatase activity (GO:0004551), regulation of nodal signaling pathway (GO:1900107), ethanol metabolic process (GO:0006067), and lipid metabolic processes (GO:00066295) (Fig. 3C, Data S4). The ELOVL fatty acid elongase 2 gene (*ELOVL2*, ENSSSCG00000001045) was found to be closely related to lipid metabolic processes (Data S4), and was found to be significantly down-regulated (log FC = −3.306051731) (Data S3). A previous study has reported *ELOVL2* as a highly expressed gene in the liver that was closely related to the elongation process of PUFAs with 22 carbons to produce 24-carbon precursors for DHA and DPAn-6 formation, indicating that it was essential for lipid homeostasis (*Pauter et al., 2014*). In addition, in testis, the ELOVL2 can also control the level of n-6 28:5 and 30:5 fatty acids, which had been identified as indispensable for normal sperm formation and fertility in male mice (*Zadravec et al., 2011*).

## Differentially expressed noncoding RNA during liver aging

To detect the expression profiling of age-related lncRNA, the identified lncRNAs were screened out during porcine liver aging. Here, 19 lncRNAs were identified as differentially expressed lncRNAs (DELs) in the aging of porcine liver. Seventeen out of these 19 lncRNAs presented up-regulation, and two exhibited down-regulation (Fig. 3B, Data S3). Additionally, GO analysis was performed to investigate the target genes of DELs bearing biological functions. The results showed that the target genes of up-regulated lncRNAs were significantly enriched for the immune response, signaling pathway, and protein transport, such as positive regulation of type I interferon production (GO:0032481), defense response to virus (GO:0051607), positive regulation of NF-kappaB transcription factor activity (GO:0051092), response to virus (GO:0009615), positive regulation of RIG-I signaling pathway (GO:1900246), epoxygenase P450 pathway (GO:0019373), and SMAD protein import into nuclei (GO:0007184) (Fig. 3D, Data S5). Moreover, the target genes of down-regulated lncRNAs were significantly related to oxidation–reduction processes, metabolic processes, diphosphatase activity, iron ion and heme binding, and transport processes, such as oxidation–reduction process (GO:0055114), oxidoreductase activity (GO:0016491), metabolic process (GO:0008152), phosphodiesterase I activity (GO:0004528), nucleotide diphosphatase activity (GO:0004551), iron ion binding (GO:0005506), heme binding (GO:0020037), lipid transport (GO:0006869), and both bile acid and bile salt transport (GO:0015721) (Fig. 3D, Data S5).

To investigate the co-regulated relationships of lncRNA-mRNA during liver aging, the DELs and their predicted target genes were screened, and three significantly up-regulated lncRNAs (*MSTRG.58269*, *MSTRG.85782*, and *MSTRG.177174*; logFC >9.0) as well as one down-regulated lncRNA (*MSTRG.205482*; logFC <-2.0) were found. Several of their

**Table 1** Gene Ontology annotations of differentially expressed lncRNA and their differentially expressed targets in OYL.

| lncRNA | Targets (corrected *p*-value) | Pearson correlation | Adjusted *p*-value | Gene Ontoloty annotations |
|---|---|---|---|---|
| OYL | | | | |
| | DMBX1 (ENSSSCG00000003900) ↑ | 0.955329954 | 5.67E−07 | DNA binding |
| MSTRG.58269 ↑ | GUCA1A (ENSSSCG00000001636) ↑ | 0.949969984 | 3.66E−06 | Calcium sensitive guanylate cyclase activator activity |
| | IRG6 (ENSSSCG00000008648) ↑ | 0.925610513 | 0.001682339 | Defense response to virus |
| | ISG15 (ENSSSCG00000027982) ↑ | 0.944967314 | 1.69E−05 | Defense response to virus |
| MSTRG.85782 ↑ | S1PR3 (ENSSSCG00000009580) ↑ | 0.926961232 | 0.001284965 | Regulation of interleukin-1 beta production |
| | IRG6 (ENSSSCG00000008648) ↑ | 0.921249274 | 0.003855634 | Negative regulation of viral genome replication |
| MSTRG.177174 ↑ | SARDH (ENSSSCG00000005740) ↑ | 0.949551978 | 4.19E−06 | Oxidation–reduction process |
| | CYP1A1 (ENSSSCG00000001906) ↓ | 0.993093224 | 1.40E−23 | Hydrogen peroxide biosynthetic process |
| | LIPK (ENSSSCG00000010442) ↓ | 0.990169559 | 5.79E−20 | Lipid metabolic process |
| MSTRG.205482 ↓ | SULT1B1 (ENSSSCG00000027194) ↓ | 0.985193868 | 4.78E−16 | Flavonoid metabolic process |
| | TLR4 (ENSSSCG00000024231) ↓ | 0.949426725 | 4.36E−06 | Lipopolysaccharide receptor activity |
| | ENPP3 (ENSSSCG00000004194) ↓ | 0.94060909 | 5.60E−05 | Nucleotide diphosphatase activity |

**Notes.**
OYL, old liver vs. young liver; ↑, Up-regulated; ↓, Down-regulated.

target genes were differentially expressed during liver aging (Table 1). Here, *MSTRG.58269* was found to have biggest change in expression (logFC = 9.880537444), and its target genes *DMBX1* (enriched for DNA binding) and *GUCA1A* (enriched for Calcium sensitive guanylate cyclase activator activity) were both up-regulated in response to liver aging. Furthermore, most of the target genes of *MSTRG.85782* were up-regulated during liver aging, and these genes were significantly enriched for immune response, such as *IRG6* and *ISG15* (enriched for the defense response to virus), *S1PR3* (regulation of interleukin-1 beta production), and *IRG6* (negative regulation of viral genome replication). In addition, the target genes of *MSTRG.205482* were down-regulated in response to liver aging, and were significantly enriched for metabolic process, such as *LIPK* (related to lipid metabolic process), *SULT1B1* (related to flavonoid metabolic process), and *TLR4* (related to lipopolysaccharide receptor activity) (Table 1).

A total of 26 differentially expressed miRNAs (DEMs) were identified in OYL, of which 14 were up-regulated and 12 were down-regulated (Fig. S1A, Data S3). Moreover, 103 differentially expressed circRNAs (DECs)were identified in OYL, of which 81 were up-regulated and 22 were down-regulated (Fig. S1B, Data S3). Here, the *miR-206* (logFC = 5.300862103) was identified to be significantly up-regulated in response to liver aging (Data S3). A previous study has shown that *miR-206* can regulate the glucokinase activity that knocked out the *miR-206* and produced the elevated glycogen content in the mice liver (*Manjula et al., 2016*). Moreover, it was found that *miR-206* represses the LXR-$\alpha$ activity and expression in hepatic cells (*Vinod et al., 2014*).

The circRNA as competitive endogenous RNA (ceRNA) regulate the function of miRNA acting as a 'sponge', thus it could regulate the expression of target genes. This indicates that circRNAs and their target miRNAs may exert a co-expression during liver aging (*Li et al., 2015*; *Sanger et al., 1976*). Here, the Pearson correlation coefficien was analyzed
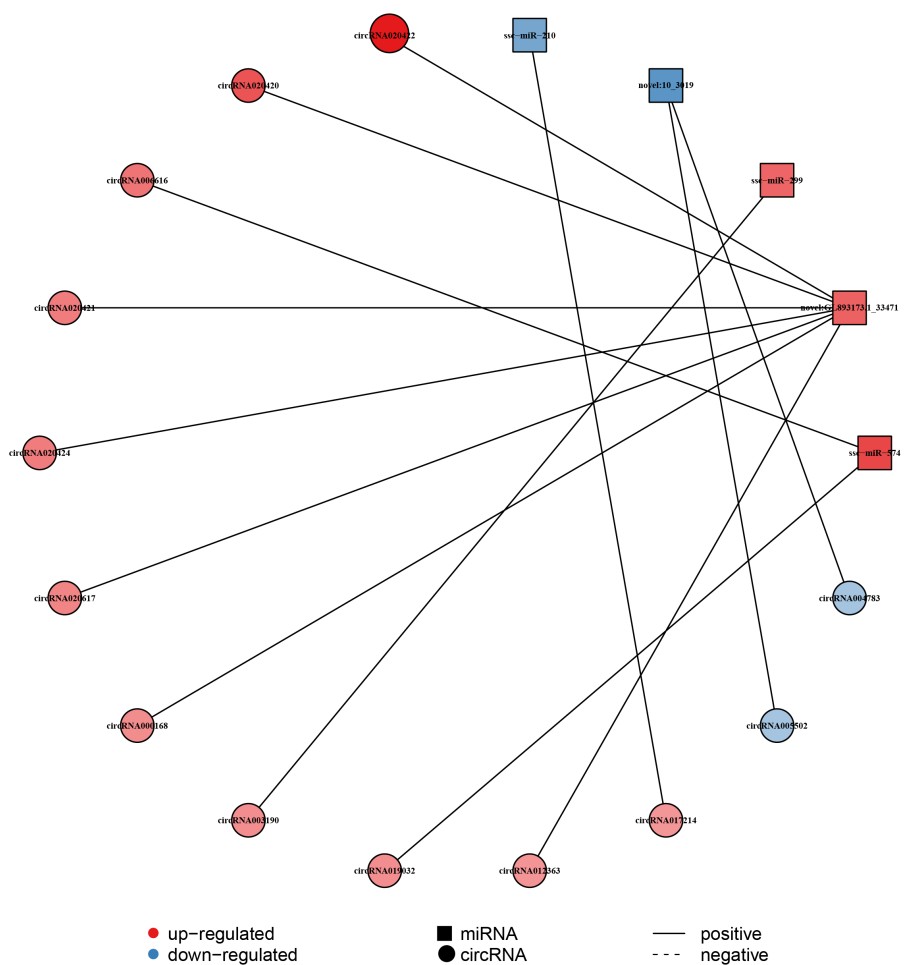

circRNA020422  ssc-miR-210
circRNA020420  novel:10_3619
circRNA000616  ssc-miR-299
circRNA020421  novel:GL893173.1_33471
circRNA024324  ssc-miR-574
circRNA020617  circRNA004783
circRNA000168  circRNA005502
circRNA003190  circRNA017214
circRNA019032  circRNA013363

● up–regulated    ■ miRNA    —— positive
● down–regulated  ● circRNA   - - - negative

**Figure 4  Construction of the circRNA-miRNA co-expression network in the liver.** The Pearson's correlation coefficient analysis was performed between the differentially expressed circRNAs and miRNAs. $r > 0.8$ and $p$-value $< 0.05$ were considered relevant for the network construction between a circRNA and a miRNA. Red color and blue color represent up and down regulation, respectively. The black solid line represents a positive correlation.

between circRNA and miRNA based on the expression levels of DECs and DEMs, and a co-expression network map was constructed (*Dou et al., 2016*). The network map included 5 miRNAs, 13 circRNAs, and 13 relationships (Fig. 4, Data S6). In this network, 11 circRNAs associated with three miRNAs were found to be synchronously up-regulated. However, two circRNAs associated with two miRNAs were found to be synchronously down-regulated. In addition, seven circRNAs with GL893173.1_33471 (novel miRNA) presented most relationships, while *circRNA003190* and *circRNA017214* showed single relationship with *miR-299* and *miR-210*, respectively (Fig. 4, Data S6).

## The expression of genes validated by q-PCR
q-PCR was performed to verify the RNA-seq results. A certain number of genes were chosen that were found to play a significant role in specific diseases or in vital biology functions.

Moreover, several DEGs were randomly selected to conduct q-PCR to determine the expression levels of these gene in the four samples (two 180-days pigs and two 8-years pigs). Selected genes were composed of two mRNAs (*ELOVL2* and *NR1H2*), two lncRNAs (*MSTRG*.6003 and *MSTRG.205482*), two miRNAs *(miR-210* and *miR-299*), and two circRNAs (*circRNA005502* and *circRNA015222*) (Fig. S2). The results showed that the expression patterns were highly consistent between the two methods ($r = 0.887581$, $p = 0.003259$) (Fig. S2).In this study, the q-PCR results affirmed the high reproducibility and reliability of the gene expression profiling from RNA-seq results. The primer pairs used here are listed in Table S3.

## DISCUSSION

In this study, we described changes of comprehensive transcriptome profiling in porcine liver during aging and a large number of mRNAs, lncRNAs, and miRNAs were found expressed in both young and old livers. However, ∼16.9% of cirRNAs were found expressed in both young and old livers, which suggests that the expression of circRNA showed a strong stage-specific pattern during aging. This may indicate that the circRNA not only regulates gene expression via competitive binding of miRNAs as ceRNAs, but is also involved in many vital roles during liver aging in mammals. Previous studies showed that circRNAs could translate to proteins and play significant roles in the inhibition of cancer development or biological functions (*Yun et al., 2017*; *Zhang et al., 2018*). This study presented that young and old individuals were separately grouped via Euclidean matrix plots on the basis of mRNA and lncRNA transcripts, and an identical result was observed on the basis of miRNA. This is in accordance with the result of the porcine skeletal and cardiac muscle aging described in a previous study by our group (*Chen et al., 2018*). These results suggest that gene expression patterns were age-dependent in liver, skeletal, and cardiac muscles. PCA showed a higher degree of variance between young livers compared to old livers based on mRNA and lncRNA transcripts. However, the PCA based on miRNA transcripts showed the opposite result i.e., a higher degree of variance between old livers compared to young livers. A previous study showed that approximately half of the total gene expression changes observed from newborns to age of 40 occurred within the first year of life in the aging process of human brain (*Somel et al., 2009*). Moreover, it has been reported that the miRNA expression in the frontal cortex appeared to be less affected by aging (*Persengiev et al., 2011*). This indicates that changes of mRNA and lncRNA expression during the liver aging process may occur earlier than those of miRNA, and aging affects the expression of miRNA relatively less in the liver.

Previous research has shown that age-related genes were involved in immune response, metabolic processes, cell activation, and RNA modification during liver aging in mice (*White et al., 2015*), and similar results were found in this study. Here, many age-related genes were identified during porcine liver aging. GO analysis showed that the up-regulated genes were mainly enriched for immune response, which indicates that the liver is more easily infected by viruses with age, and the risks for various liver diseases also increased with age (*Kim, Kisseleva & Brenner, 2015*). However, down-regulated genes were mainly

involved in metabolism, which may be related to the steady decline of the metabolic ability of the liver (the largest metabolic organ) in response to age. Here, the *ELOVL2* gene was found to be significantly down-regulated during porcine liver aging, and it was significantly enriched for the lipid metabolic process. More importantly, previous studies have found that the DNA methylation of the *ELOVL2* gene correlated strongly with age in human liver (*Bacalini et al., 2018*; *Bysani et al., 2016*). In addition, the *ELOVL2* gene also has been found to undergo age-associated hypermethylation on the CpG island in other tissues (*Bacalini et al., 2018*), and displays the widest methylation range in response to age in whole blood and white blood cells (*Bysani et al., 2016*). These results suggest that the reason why the *ELOVL2* gene was down-regulated may be closely related to the declining metabolic ability of the liver with age. Furthermore, the hypermethylation of the CpG island of the *ELOVL2* gene may contribute to regulating the *ELVOL2* expression in liver aging. Therefore, the *ELOVL2* gene may be an important biomarker during liver aging in humans and big mammals. Moreover, the results of this study suggest that several aging-related gene changes may be related to a decline or alteration of the overall function of liver organs. This is a reasonable suggestion based on the obtained results. However, no direct evidence exists that there is a direct relationship between both; therefore, this need further extensive functional validations.

Moreover, the nuclear receptor subfamily 1 group H member 3 (NR1H3, ENSSSCG00000013241), encoding LXR-$\alpha$, has been reported to be highly expressed in four samples, and it has been reported that the LXR-$\alpha$ activity and its expression could be repressed by *miR-206* in hepatic cells (*Vinod et al., 2014*). More importantly, a previous study has shown that LXRs as lipid-dependent regulators of inflammatory gene expression may serve to link the lipid metabolism and immune functions in macrophages (*Joseph et al., 2003*). Interestingly, *miR-206* was also identified to be up-regulated during porcine liver aging, and several up-regulated genes were also enriched for the immune response in this study. Consequently, this indicates a contribution why the *NR1H3* was not increased like the *NR1H2* during porcine liver aging. In addition, the gene *NR1H2* was identified to be up-regulated in the liver with increasing age, and the *NR1H2* encodes LXR-$\beta$, which is universally expressed (*Kalaany & Mangelsdorf, 2006*). A deficient glucose-dependent insulin secretion and an enhanced lipid accumulation have been reported in LXR-$\beta$ knockout mice (*Gerin et al., 2005*). Furthermore, the variation within the *NR1H2* gene may facilitate the development of type 2 diabetes (*Ketterer et al., 2011*). It has been reported that pigs have a genomic structure, as well as physiological and biochemical features that are similar to those of humans (*Welsh et al., 2009*); however, pigs cannot develop type 2 diabetes. The obtained results indicate that the increased *NR1H2* expression in the aging liver may be one of the key effects involved in the resistance to the development of type 2 diabetes.

In this study, the research animal is the Yanan pig, which has been continuously raised from long ago in the hilly regions of the Sichuan Province. During the last two decades, the production of The Yanan pig has been reduced due to its poor carcass composition and growth performance compared to the white pigs and thus it is believed that the Yanan pig is an endangered pig breed (*Jiang et al., 2012*). For this reason, since 2014, the Yanan pig has

been included in the National Program for Farm Animal Resources (*Yan et al., 2018*). Until now, only approximately 100 Yanan pigs have been protected in the conservation farm. Here, 180 days and 8 years were selected as the two time points to investigate liver aging, and the investigated sows were healthy and normally growing pigs. The two 180-day-old sows had not previously given birth, while the two 8-year-old sows were in the depleted stage of production performance after many years of giving birth and were eliminated; this enabled the inclusion of two 8-years sows in this research. Although this research offers a meaningful reference for the study of the liver aging process between 14 and 50 years of age in humans, the aging process of the porcine liver needs to be studied at a much older age as it has been shown that the aging rate of the human liver accelerates above the age of 50 (*Kudryavtsev et al., 1993*). However, 8-years old sows were the oldest available pigs because older sows have been eliminated before due to their depleted production performance. In addition, the Yanan pig is a very treasured pig breed and their population size is very small, which further limited the selection of a higher number of samples in each group for the study of the senescence of pig liver, especially for 8-year-old sows.

## CONCLUSIONS

Aging is a key factor for the development of a large number of diseases, and it can significantly influence the most important metabolic organ the 'liver'. Here, RNA-seq was conducted to determine the expression profiles of mRNA, lncRNAs, miRNAs, and circRNAs. Gene function enrichment analysis showed that up-regulated and down-regulated genes were mainly involved in immune response and metabolism, respectively. A certain number of lncRNAs and their target genes were both observed to be differentially expressed during liver aging. circRNA-miRNA co-expression networks were constructed during liver aging. The obtained findings contribute to the understanding of the transcriptional basis of liver aging and also provide further references for the understanding of human liver aging at the molecular level.

### Funding

This study was supported by the National Key R&D Program of China (2018YFD0501204), Key Program of National Science Foundation of China (Grant No. 31530073), and Chengdu Local Good Varieties of Livestock and Poultry Resources Protection and Exploitation and Utilization of Construction Projects. The funders had no role in study design, data collection and analysis, decision to publish, or preparation of the manuscript.

### Grant Disclosures

The following grant information was disclosed by the authors:
National Key R&D Program of China: 2018YFD0501204.
Key Program of National Science Foundation of China: 31530073.
Chengdu Local Good Varieties of Livestock and Poultry Resources Protection and Exploitation and Utilization of Construction Projects.

## Competing Interests

Daojun Lv and Peilin Li are employed by Sichuan Weimu Modern Agricultural Science and Technology Co., Ltd.

## Author Contributions

- Jianning Chen conceived and designed the experiments, analyzed the data, contributed reagents/materials/analysis tools, prepared figures and/or tables, authored or reviewed drafts of the paper, approved the final draft.
- Qin Zou conceived and designed the experiments, performed the experiments, analyzed the data, contributed reagents/materials/analysis tools, prepared figures and/or tables, approved the final draft.
- Daojun Lv, Xue Wang, Yan Chen, Xiaoyu Xi, Peilin Li and Anxiang Wen performed the experiments, approved the final draft.
- Muhammad Ali Raza performed the experiments, authored or reviewed drafts of the paper, approved the final draft.
- Li Zhu, Guoqing Tang, Mingzhou Li and Xuewei Li contributed reagents/materials/-analysis tools, approved the final draft.
- Yanzhi Jiang conceived and designed the experiments, analyzed the data, contributed reagents/materials/analysis tools, prepared figures and/or tables, authored or reviewed drafts of the paper, approved the final draft.

## Animal Ethics

The following information was supplied relating to ethical approvals (i.e., approving body and any reference numbers):

All experiments involving animals were conducted according to the Regulations for the Administration of Affairs Concerning Experimental Animals (Ministry of Science and Technology, China, revised in June 2004), and approved by the Institutional Animal Care and Use Committee in College of Animal Science and Technology, Sichuan Agricultural University (Sichuan, China) under permit No. SKY-B20160201.

## Data Availability

The RNA-seq data are available at the NCBI Gene Expression Omnibus (GEO) under the accession GSE123590.

## Supplemental Information

Supplemental information for this article can be found online at http://dx.doi.org/10.7717/peerj.6949#supplemental-information.

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
