# Peer review of "Comprehensive transcriptional profiling of aging porcine liver"

_PeerJ, doi:10.7717/peerj.6949_

## Round 0.1 · original submission · Minor Revisions

Please address the reviewers' concerns point-by-point.

Reviewer 1 ·

Basic reporting

The manuscript entitled “comprehensive transcriptional profiling of porcine live aging” describes the basis analysis of RNA-seq data obtained from the porcine liver. More in-depth analysis would be appreciated. In addition, the grammar and interpretation need more work to make it an easier read. For example, it is hard to make sense of line 251-254.

Experimental design

qRT-PCR validation should be described within subsection of the type of RNA analyzed, not as a separated subsection and bundled together. Also should be presented as main figures, not supplemental data.

Validity of the findings

1. While the authors emphasize on the “understand the transcriptional regulatory mechanism of the liver aging”, work presented is really just basic analysis of RNA-seq data. Therefore, some sentences in the Introduction and Discussion need to be toned down.
2. Should it be differential expressed, not co-expressed (line 251)?

Additional comments

The font of figure label need to be increased,

Reviewer 2 ·

Basic reporting

(1) There are some problems with sentence structure, verb tense, and clause construction. There are several typographical and grammatical mistakes in the paper, which needs to be proof read carefully and checked. The English of your manuscript must be improved before re-submission. For example, the font of phrase “piglets weaning was performed” at line 79 is not the same as other characters; “OL” should be revised with “OYL” at line 224 and 226; “compared to” should be revised with “between” at line 255; the title of figure legends should be revised to avoid the grammatical mistakes; “NR1H2” should be used the Italic font at line 276 and 180-186 etc. It is strongly suggested that the authors obtain assistance from a colleague who is well-versed in English or whose native language is English.

(2) Literature references should be added up several latest reports in parts of Introduction and Discussion. What are the specific representations of senescence associated liver pathological changes and dysfunctions? How are the above changes related to the results of the transcriptional profiling you reported? Please add up the above literature references and discussion in the part of Discussion. It is advised that the phrase “Compared to the young group” should be added up into the line 176.

(3) The paragraphs from line 212 to 239 could be an individual part and named “The cross-talk between mRNA and non-coding RNA during liver aging”. Statistical analysis should be showed in the part of Materials & Methods as an individual paragraph. SD or SEM and P-values were not showed in the results of qPCR in Figure S2.

Experimental design

It is advised that liver senescence and associated pathological changes and dysfunctions would be detected and analyzed using histology, immunity and metabolism testing to consist with the results of transcriptional profiling. Moreover, please offer the Permit Number on the ethics of animal experiments by the local ethics committee.

Validity of the findings

Statistical analysis should be showed in the part of Materials & Methods as an individual paragraph. Please added up the detail of statistical analysis in each figure legend and showed the SD or SEM and P-values in the figures.

Additional comments

The authors have provided evidences to support the transcriptional profiling of porcine liver aging. Some senescence associated genes of porcine liver were found by RNA sequencing and analyzed by GO annotation. Non-coding RNA were also reported and assessed the cross-talk with the above related target genes. The experiment design and data are sound, but the manuscript is far from qualified to publish in Peer J now. There are some suggestions above that the authors should address to help improve the quality of the manuscript.

Reviewer 3 ·

Basic reporting

The language should be improved in parts to more clearly convey the author's intended meaning. Some phrasing and sentence structures make comprehension difficult, for example, lines 45-48, 50-52, 62-67 etc. Shorter and clearer sentences will help.

The figures and tables are neat and contain the requisite explanations.

Experimental design

Methods:
- please provide reference for the feed used for the young pigs. Is this the standard feeding regimen?

- The history/conditions of the old pigs was not discussed at all

- Why were 180 days and 8 years selected as the two time points? Please explain the rationale with respect to the normal lifespan of a pig.

- Were any tests performed to determine the health status of the older pigs? This is relevant information considering the results discuss the up-regulation of viral associated immune genes.

- Please mention the version of pig reference genome used in methods section (it is noted only in results)

- Please mention what samples were used for the qPCR? Were they the same ones used for the transcriptome analysis?

Validity of the findings

Results and Discussion:

- The sample size would ideally be 3 samples each for young and old livers but the study only has two which weakens the statistics. Also, it is mentioned that expression of mRNA is counted even if only one of the samples contains expression. This further weakens the significance of the statistics and can potentially skew the PCA plots. Since there are only two samples for each condition, the authors might restrict inclusion of expressed mRNAs only to those present in both samples and re-test the heatmap and PCAs.

- Please explain why mRNA and lncRNA expression was grouped to generate Fig 2A

- The results have been reported well but some of the inferences may be a little weak just with the observed results alone. For example, down-regulation of some metabolism related genes may not necessarily mean it is proof of compromised metabolic activity at an organ level. Authors should acknowledge that further extensive functional validations are required to claim with certainty.

- Validation by qPCR should be performed on multiple independent samples to confirm the expression pattern as seen by transcriptome analysis. This becomes more important since the initial number of samples is only two.

Additional comments

The study seeks to answer an interesting question which is not covered in the literature. The data sets presented can be useful to other researchers. In order to increase utility, you can provide ranked lists of DEGs derived from your raw data in the form of tables.

Overall the manuscript needs work on some sentence structure and phrasing. Simpler sentences will help adding clarity to the reader.

---

## Round 0.2 · accepted · Accept

Thank you for appropriately revising the manuscript.

#